# Perennial Grasses on Stony Sandy Loam *Arenosol*: Summary of Results of Long-Term Experiment in Northern Europe Region (1995–2024)

**DOI:** 10.3390/plants14020166

**Published:** 2025-01-09

**Authors:** Liudmila Tripolskaja, Asta Kazlauskaite-Jadzevice, Almantas Razukas, Eugenija Baksiene

**Affiliations:** Voke Branch, Lithuanian Research Centre for Agriculture and Forestry, Zalioji 2, LT-02232 Vilnius, Lithuania; almantas.razukas@lammc.lt (A.R.); eugenija.baksiene@lammc.lt (E.B.)

**Keywords:** grassland, biomass productivity, species diversity, SOC, marginal soil

## Abstract

Grasses can sustain soil functions despite nutrient depletion, which can have serious consequences for soil processes and ecosystem services. This paper summarizes the results of the long-term experiment (1995–2024) carried out in *Arenosol* within a temperate climate zone, focusing on the productivity of natural and managed grasslands; their succession changes over time, and so do the effects on soil chemical properties, and soil organic carbon (SOC) sequestration. The results indicated that two land uses—abandoned land (AL) and grassland fertilized with mineral fertilizers (MGf)—can be effectively applied to prevent *Arenosol* soil degradation. SOC accumulation occurs more rapidly in AL soils, and their chemical properties show less change over time. The ability of grasses to sequester SOC is better reflected by SOC stocks across the Ah horizon, where thickness varies over long-term grassland use. Significant changes in soil properties were observed more than 20 years after converting arable to herbaceous land use. While MGf has the highest biomass productivity, the use of fertilizers leads to soil acidification. The biomass productivity of AL and MGf increased with longer grassland use; however, in MG, productivity decreased without fertilizers, reaching AL’s productivity levels after 20 years. As the age of AL increased, plant biodiversity decreased, and drought-resistant plants began to spread.

## 1. Introduction

Perennial agricultural grasslands and natural and semi-natural grasslands occupy about 34% of Europe [1]. They are widely used for livestock production and are also an effective means of preventing soil degradation. Grass can sustain soil functions despite nutrient depletion, which can have serious consequences for soil processes and ecosystem services. In recent decades, the ability of herbaceous land to accumulate carbon in the soil more rapidly than other land uses has been particularly emphasized. Consequently, the aim is to expand these areas to reduce the impact of carbon dioxide (CO_2_) on climate change and improve soil health [2,3,4]. At the same time, grasslands perform other ecosystem services important for society’s needs, which include the preservation of landscape and aesthetic values, biodiversity of flora and fauna, etc. [5,6]. The European Union’s (EU) agricultural policy running from 2023 to 2027 increases the ambitions of the European Green Deal by replacing current greening systems and increasing incentives for climate and environmentally friendly agricultural practices, encouraging an increase in grassland use areas, as they are more active in sequestering organic carbon and have a wider biodiversity.

In regions with extensive areas of marginal soils, the use of grasslands is significant for restoring soil fertility and preventing its degradation. This is especially relevant for soils with a light texture (sand, loamy sand), which are characterized by an insufficient amount of nutrients, low sorption capacity, and water absorption. The use of such soil as cropland usually has a negative effect on soil fertility, because the amount of organic carbon in it often decreases due to applied agrotechnical measures, and as a result, the physical and biological properties of the soil deteriorate [7,8,9]. In one way, applying all scientifically based agrotechnical measures can improve soil properties in arable land use in such soils as well [10].

In the context of organic carbon sequestration, the superiority of grassland use compared to cropland is confirmed by many scientific experiments [11,12]. Other scientists [13], after performing the quantification of current soil organic carbon (SOC) stocks in agricultural systems in the EU between 2009 and 2018, found that cropland cultivation was the factor that contributed to the lowest negative change in Δ SOC with an estimated partial effect of −0.04  ±  0.01 g SOC kg^−1^ year^−1^, while grassland caused the highest positive change with an estimated partial effect of 0.49  ±  0.02 g SOC kg^−1^ year^−1^. This confirms the SOC sequestration potential of converting cropland to grassland. However, it is important to consider that local soil and environmental conditions may either diminish or enhance the grassland’s positive effect on soil SOC storage.

The effect of herbaceous land use on SOC sequestration depends on many factors: soil properties, herbaceous botanical composition, age, use practices, fertilization, water regime, climatic conditions, and others [14,15,16,17,18]. According to [19], optimal management methods should be adapted to regional and local circumstances, considering a variety of factors.

As the climate changes, it becomes important to assess the impact of the increasing average annual air temperature on SOC mineralization and sequestration. New experiments reveal that organic matter degradation in soil occurs faster as air temperature rises, and this needs to be evaluated when predicting the rate of OC sequestration in both herbaceous and other land uses [17,20]. We believe that long-term experiments help in more objectively assessing the impact of land uses on soil fertility, transforming regularities of grasslands depending on their use practices and age, and enabling changes in soil properties to be tracked, such as carbon sequestration dynamics over a long period of time.

This paper summarizes the results of the long-term experiment (1995–2024) carried out in *Arenosol* within a temperate climate zone, focusing on the productivity of natural and managed grasslands, their succession changes over time, the effects on soil chemical properties, and SOC sequestration. We believe that the established patterns will enable a more precise selection of agrotechnical practices, facilitating a harmonious combination of EU Green Deal aspirations and socio-economic needs in regions with marginal soils.

Land use must prioritize the efficient production of biomass to achieve better economic, environmental, and social results. Our focus should shift toward integrated, systems-based approaches to land management that promote the sustainable intensification of agricultural production, even in areas such as neglected grasslands and abandoned lands. These areas share the common issue of having lost their economic and/or ecological viability for the community. This challenge is exacerbated by the ongoing degradation of such lands, which leads to a decline in their capacity to provide essential ecosystem services.

## 2. Materials and Methods

### 2.1. Experiment Site Description and Field Investigation

The experiment was established in the Voke branch of the Lithuanian Research Centre for Agriculture and Forestry (Vilnius, Lithuania, 54°33′49.8″ N 25°05′12.9″ E) in 1995. The experiment location is in the temperate climate zone and belongs to the Nord European region (Figure 1).

The experimental site is located in the low-productivity *Endocalcaric Arenosol*, which was formed on fluvioglacial deposits. Its soil profile consists of the following horizons: Ah-AB-B1-B2-2Cα1-2Cα2 (Figure 1). The soil has a light texture, and according to the amount of sand and clay particles (sand 63 μm–2 mm—81.0–83.7%; silt 2–63 μm—11.2–13.7%; and clay < 2 μm—4.5–5.0%), it is classified as loamy sand [21] (Table 1). *Arenosols* formed by fluvioglacial deposits are characterized by a significant number of stones throughout the profile, which increases in the lower layers of soils. The stoniness of the soil horizons in the 0–15 cm layer was, on average, 15.2–17.7%; 15–30 cm—18.6–20.1%; and 30–50 cm—22.9–35.2%.

The purpose of the conducted experiment was to determine the most suitable conversion methods of arable land use to other types of land use (managed grassland, abandoned land, pine plantation) in barren sandy soils. This paper examines only the influence of herbaceous land use (MG and AL) on soil properties and the change in their productivity and plant succession over the course of 30 years of research. The cultural grassland site was divided into 2 parts—grassland fertilized with mineral fertilizers (MGf) and nonfertilized (MGunf). The dynamics of the development of natural vegetation succession and the change in its productivity were studied in the abandoned land site (AL). The areas of land use under investigation were 400 m^2^ for AL and 200 m^2^ for MGunf and MGf.

The herbage of the MG site consisted of plants of the *Legume* and *Poaceae* families. The seed mixture consisted of 40% hybrid alfalfa (*Medicago varia* L.) and 4 species of grasses—20% red fescue (*Festuca rubra* L.), 20% bromegrass (*Bromus inermis* Leyss.), 10% cock’s-foot grass (*Dactylis glomerata* L.) and 10% meadow-grass (*Poa pratensis* L.). The same grass mixture was grown in 2007–2024, but after 2007, cock’s-foot was replaced with timothy (*Phleum pratense* L.). As the grass matured, it was reseeded every 10 years (in 2004 and 2015) to improve its fodder value and productivity. At the MGf site, the grassland was fertilized with mineral fertilizers each spring and again after the first grass cutting. In 1995–2015, 60 + 30 kg ha^−1^ N (ammonium nitrate), 39 kg ha^−1^ P (granular superphosphate) and 100 kg ha^−1^ K (potassium chloride) were applied. After 2016, after noticing a trend of decreasing available potassium at the MGf site, the amount of potassium fertilizer was increased to 125 kg ha^−1^ K. Grasses were harvested 2 times during the vegetation period. The first grass cutting was performed during the flowering phase of alfalfa (in the first ten days of July). The second cutting was performed in the first ten days of September. The biomass of the MG site was removed from the experimental area.

In 1995, on the AL site, plant cover began to form from the seed bank in the soil. No agrotechnical measures were used on this site. At the end of July, the site vegetation was cut to prevent the spread of bushes and trees on the site. The cut plant biomass was left on the soil surface.

### 2.2. Calculation of Biomass Yield

Above-ground grass biomass was calculated in the MG and AL sites each year. The dry matter content in plants was determined after the samples had been dried to constant moisture at 105 °C. Biomass calculation in the AL site was performed at 4 spots of 0.25 m^2^, and in the MGf and MGunf plots, the yield calculation was performed for 48 m^2^ of recorded subplots with 3 replications. Dry matter yield (DM Mg ha^−1^) was determined.

The analysis of the species composition of plants in the experimental site was carried out in 1995, 2000, 2004, 2015, and 2022. The vegetation cover was estimated according to Braun–Blanquet’s method [22]. Identification of plant species was carried out using the K.K. Vilkonis atlas [23].

### 2.3. Soil Sampling

This paper presents indicators of soil chemical properties in 1995, 2018, and 2024, which illustrate the dynamics of their change over a period of 30 years. Samples were taken from the 0–25 cm layer in 3 replicates. The soil samples for the analytical determinations were homogenized and air-dried before being gently crushed and passed through a 2 mm mesh sieve. During the soil sample preparation, coarse material (roots and litter) was removed.

Prior to soil sampling, profiles were excavated at three locations in each type of land use (in a 20 m × 20 m grid), to a depth of 50 cm, in order to accurately determine the depths of the A and AB horizons in each type of land use, and to determine the soil bulk density and the soil chemical properties. The soil bulk density (Ah horizon) was determined by the core method, using a metal ring pressed into the soil (intact core) and determining the weight after drying [24]. Core cutter samples were taken in three replicates.

The soil chemical properties were determined as follows:pH_KCl_: by the potentiometric method (1 mol L^−1^ KCl using a soil/solution ratio of 1:2.5), ISO 10390:2005 [25];Plant available P_2_O_5_ and K_2_O were extracted using 0.03 M ammonium lactate (Egner–Riehm–Domingo (A–L) method);SOC concentration: by the Duma method (after dry combustion), ISO 10694:1999 [26].

Soil organic carbon stocks in the A horizon were calculated as follows [27]:SOCstock (Mg ha^−1^) = SOCcon × BD × depth/10

where SOCcon is the soil organic carbon concentration (g kg^−1^), BD is the bulk density of the Ah horizon (Mg m^3^), depth is the thickness of the humic Ah horizon layer (cm), and 10 is the coefficient used to calculate the SOC stocks in Mg ha^−1^.

### 2.4. Meteorological Conditions

The location of the experiment belongs to the Nord European region and according to the Köppen climate classification [28]; the climate of the study area is classified as Dfb—humid continental with warm summers, but rather cold winter periods. In recent decades, the climate in Lithuania has changed. After calculating the new standard climate norm for precipitation and air temperature, it was found that the standard precipitation norm (SCN) for 1991–2020 was 695 mm, an increase of 31 mm compared to the 1961–1990 SCN, and the average annual air temperature increased from 6.4 to 7.4 °C [https://www.meteo.lt/app/uploads/2023/11/standartine-klimato-norma-1991-2020.pdf, accessed on 5 January 2025].

During the research period, the annual amount of precipitation varied from 519 mm in 1996 to 963 mm in 2010. The rainiest years (precipitation > 800 mm) were 2010, 2011, and 2017, while the driest years (with precipitation < 600 mm) were 1996, 1999, 2018, 2019, 2020, and 2023 (Figure 2). Annual rainfall has notably decreased since 2018 and was significantly lower compared to SCN. Although compared to the 1961–1990 precipitation SCN, the 1991–2020 SCN increased, the trend of the change in precipitation did not show a significant increase during the research period (r = 0.044).

The average annual air temperature was also very different in individual years of the experiment, varying from 5.3 °C in 1996 to 9.0 °C in 2020. The years 1996, 1997, 2003, 2004, and 2010 were cooler (≤6.5 °C), and 2000, 2015, 2018, 2019, 2020 and 2023 were warmer (≥8.0 °C). The trend in temperature changes indicates an increasing tendency (r = 0.62, slope = 0.062) throughout the course of the experiment (Figure 3).

### 2.5. Statistical Analysis

The data were structured and analyzed using the Microsoft Excel software package and the SAS Enterprise software, version 7.1 (SAS Institute Inc., Cary, NC, USA). The differences between the experimental treatments were tested using a one-way analysis of variance (ANOVA), using Fisher’s post hoc test to determine the least significant difference between treatments. The significance of differences was set if the probability level was lower than or equal to 0.05. A leaner and polynomial regression analysis was used to reveal the relationship between biomass DM yield and the duration of the experiment, and between biomass DM yield and the number of plant species. Duncan’s post hoc test was used to carry out multiple comparisons.

## 3. Results

### 3.1. Impact of Grassland Management on Soil Humic Horizon Properties

*Arenosol* is classified as marginal soils because, due to the large amount of sand particles, they usually have little organic carbon, plant nutrients, poor water absorption, and other physical properties.The experimental field was established in 1995 on land previously used for arable farming, where mineral fertilizers were used for a long time, so the Ah horizon of the soil was moderately rich in mobile phosphorus (69–77 mg kg^−1^ P) and potassium (141–144 mg kg^−1^ K) (Table 2). The soil reaction was neutral or close to neutral (pH_KCl_ 6.0–6.8). This was caused by a highly subsiding (about 80 cm deep) layer of carbonate rock with fluvioglacial deposits. The humic horizon of the soil at the beginning of the experiment (1995) was 0.28 m thick and had little organic carbon (9.9–10.2 g kg^−1^ SOC) (Table 2).

Over the course of the experiment, the properties of the Ah horizon changed depending on the grassland management practices. With the continued use of grassland for fodder production and mineral fertilization (MGf), the concentration of available phosphorus in the soil gradually increased over the research period, rising from 77 to 119 kg kg^−1^ P. This shows that the amounts of phosphorus fertilizers ensured the nutritional needs of plants and allowed the amount of available phosphorus in the soil to increase. The amount of available potassium began to decrease slightly after 20 years of herbage use, which may limit plant biomass yield. Therefore, since 2016, the amount of potassium fertilizer applied annually has increased from 97 to 125 kg ha^−1^ K. This amount of fertilizer ensured a stable amount of available potassium in the soil. When grassland is used for fodder production without the application of mineral fertilizers, the amount of available nutrients in the Ah horizon decreases. During the first 20 years at the MGunf site, the amount of available phosphorus decreased significantly (−40 mg kg^−1^ P, *p* < 0.05); after 2018, the changes were insignificant. Changes in available potassium in the soil were similar. From 1995 to 2018, its concentration decreased by 86 mg kg^−1^ K; afterward, the decrease stabilized and the changes were insignificant. In other words, without restoring the mineral elements removed with the plant biomass, the soil became even less fertile, compared to its initial state at the beginning of the experiment installation. At the AL site, the chemical properties of the soil have changed slightly over 30 years. The biomass of the plants grown on this site remains on the soil surface and after its mineralization, the mineral elements return to the upper soil layer. After 20 years, a decreasing trend of available potassium concentration was observed in the soil at the AL site, because potassium easily leaches out of the upper soil layer in sandy and loamy soils in temperate climate zones.

During the period 1995–2018, soil acidification took place in the MGf and MGunf sites. During the first 20 years, pH_KCl_ decreased by 1.2 and 0.8 units, respectively. From 2018 to 2024, the acidification rate decreased and was statistically insignificant compared with 2018. At the AL site, soil acidity did not change significantly during the period of the experiment.

Cultivation of grass promoted an increase in Ah horizon thickness and SOC concentration. During the first 20 years of the experiment, the thickness of the Ah horizon increased the most at the AL site (+3.0 cm), less at the MGf site (+1.3 cm), and least at the MGunf site (+1.0 cm), but this change was statistically insignificant (*p* > 0.05) (Table 3). In the period 2018–2024, the thickness of the Ah horizon in all studied land uses increased slightly, and only in AL was the change significant compared to 1995 (+4.3 cm, *p* < 0.05).

Throughout the research period, the amount of SOC in the 0–25 cm layer gradually increased at the MGf and AL sites, but after 30 years, its amount increased significantly only in the AL site (+2.4 g kg^−1^ SOC). At the MGunf site, SOC changes during the entire study period were insignificant.

In the 2018 experiment, more detailed studies of SOC concentration changes throughout the Ah horizon were also carried out, in which its concentration was determined in the 0–20 cm layer and from 20 cm to the end of the Ah horizon, which made it possible to calculate SOC accumulations throughout the Ah horizon [29]. Determining the concentration of SOC at individual depths of the Ah horizon allowed for a more accurate assessment of the effect of grasses on SOC sequestration in the soil. Evaluating the changes in the parameters mentioned, it was estimated that, compared to the beginning of the research, SOC accumulation in the humic horizon increased significantly across all grasslands investigated (*p* < 0.05) (Figure 4). In 1995, SOC accumulations were 38.0–40.2 Mg ha^−1^, which increased to 44.0–48.5 Mg ha^−1^ in 2018. At the MGf and AL sites, the amount of SOC increased very similarly (7.7–10.5 Mg ha^−1^); at the MGunf site, the increase was the smallest (4.9 Mg ha^−1^), but the differences between different grasslands were statistically insignificant (*p* > 0.05).

### 3.2. Impact of Grassland Management on Biomass Yield

Throughout the experiment, the biomass yield of grasslands with different uses varied depending on grassland purpose (e.g., abandoned land or managed grassland), mineral fertilizer application, meteorological conditions, and other factors (Figure 5). In 1996–2024, the trends of biomass yield variation show that it tended to increase at the MGf site (r = 0.46). The same increasing trend was found at the AL site (r = 0.69). At the MGunf site, as the period of use of the grassland increased, the yield of grass biomass tended to decrease, which may be related to the lack of nutrients in the soil.

Depending on the meteorological factors, the DM yield of grass varied from 2.25 Mg ha^−1^ to 12.55 Mg ha^−1^ at the MGf site, from 1.1 to 7.03 Mg ha^−1^ at the MGunf site, and from 0.78 to 4.22 Mg ha^−1^ at the AL site. At the AL site, grass biomass yield had the lowest variation (VarS = 0.79), which means that natural vegetation was more resistant to the effects of environmental factors compared to cultivated grasses. The highest yield variation (VarS = 6.84) was characteristic of the MGf site for grass.

The impact of environmental factors (biotic and abiotic) on the change in biomass yield was compared between 1996–2003 and 2017–2024 average DM yields during the period. These periods include the biomass yields of MGf and MGunf grasses in the 2–9th year of use, because in the first growth year after sowing, the grasses form a smaller yield. In the first period (1996–2003), the lowest biomass yield was formed at the AL site (Table 4). It was 357% lower compared to MGf, and 279% lower compared to MGunf (*p* < 0.05).

When no fertilizer was used in the MG site, the yield of grass biomass was lower compared to the fertilized, but in the first 10 years, the difference was statistically insignificant (*p* > 0.05). Another 15 years later and with significant changes in soil properties, the highest biomass yield was also formed at the MGf site. It was significantly higher compared to the average yield from 1996 to 2003 at the MGf site (*p* < 0.05) and higher than at the AL and MGunf sites (*p* < 0.05). Without the use of fertilizers in the MG site, the biomass yield of grasses decreased significantly after 20 years, becoming similar to that of the AL site (*p* > 0.05). It should be noted that although there was a large variety of plants in the AL site (varied from 9 to 17 plant families, 25–39 species), the plants of the *Poaceae* had a greater influence on the biomass yield, whose above-ground biomass was 28–52% of the mass of all plants (Figure 6).

Evaluating the entire research period, it can be concluded that natural vegetation at the AL site produces the lowest grass biomass. Its average biomass yield was 296% lower compared to the MGf site and 156% lower than the MGunf site. This testifies to the lower ability of natural vegetation to fix CO_2_ during photosynthesis, compared to cultivated grasses.

### 3.3. Differences in Botanical Composition of Managed Grassland and Abandoned Land

At the MG site, the grass was formed from a mixture of *Legume* (*Medicago varia* L.) and four species of *Poaceae* grasses. To have a protein-rich grass feed, the mixture contained 40% alfalfa seeds. Therefore, in the first year of sowing, alfalfa plants dominated the grass mixture; their DM biomass was 81.1–91.6% of other plants (Figure 7). In the first year of grass growth, cultivated grass plants prevailed in the mixture and there were almost no plant species of natural vegetation; they constituted only 0.1% of the total biomass. By the second year of cultivation, the proportion of alfalfa in the mixture decreased to 41.1–50.0%, which was similar to that in both the fertilized and unfertilized grasslands. As the grass aged (in the fourth to fifth year of growth), the negative effect of nitrogen fertilizers on alfalfa development became more apparent. At the MGf site, the alfalfa proportion was only 7%, while at the unfertilized site, it ranged from 24.6% to 28.0% that year. The share of *Poaceae* grasses in the fertilized site increased to 57.6–99.3%, and this proportion dominated even in the later years of cultivation. At the nonfertilized site, *Poaceae* plants began to dominate from the seventh year of grass growth, with their share in the mixture increased to 61.8–89.1%. The share of natural plants in the phytocenosis of grasses was small in both the fertilized and nonfertilized sites, and the most common represented up to 5% of the total biomass, except in isolated cases. As the grassland aged, the proportion of various herbs in the mixture of cultural plants changed insignificantly.

In 1995, at the AL site, vegetation began to form from the seed bank that had been in the soil. In the first year, 39 different types of plants belonging to 17 botanical families grew in the phytocenosis of the site (Figure 8), including mayweed (*Tripleurospermum inodorum* L.), pennycress (*Thlaspi arvense* L.), pansy (*Viola arvensis* Murray.), cornflower (*Centaurea cyanus* L.), shepherd’s purse (*Capsella bursa pastoris* (L.) Medik.), goosefoot (*Chenopodium album* L.), amaranth (*Amaranthus retroflexus* L.), foxtail (*Setaria viridis* L.), and couch grass (*Elytrigia repens* L.). After the first year of abandonment, *Elytrigia repens* of the *Poaceae* family was the dominant species in the community. It is typical of arable land use in stony sandy soils of plant species.

After analyzing the botanical composition of the soil after 6 years, it was found that the diversity of plants decreased: 25 plant species from 10 families were growing on the site. Ten years after the formation of the soil (1995–2004), the diversity of plants increased again. In 2004, 33 different plant species from 14 families grew there. Plants of three families dominated: *Asteraceae*, *Fabaceae*, and *Poaceae*. After 10 years of abandonment, perennial plant species dominated, accounting for 79% of all plant species.

In 2015, a decrease in plant diversity was again detected in the soil. That year, only 18 species of plants from 11 families grew there. As in previous years, three families of plants—*Asteraceae*, *Fabaceae* and *Poaceae*—prevailed. Common yarrow (*Achillea millefolium* L.), mouse-ear hawkweed (*Pilosella officinarum* F.W. Schultz et Sch. Bip.), and species of cultural perennial grasses—tall fescue (*Festuca arundinacea* Schreb.) and orchard grass (*Dactylis glomerata* L.)—began to spread in the soil. After 20 years of abandonment, perennial couch grass (*Elytrigia repens* L.) disappeared, or its coverage dropped significantly. The coverage of bird’s-foot trefoil (*Lotus corniculatus* L.) also dropped. Yarrow (*Achillea millefolium* L.) and mouse-ear hawkweed (*Pilosella officinarum* F.W. Schultz et Sch. Bip.) were perennial species that showed continuous spread. These small herbaceous plant communities usually settle in grasslands, pine forest sites, and woodlands.

Further research conducted in 2022 showed that further changes in plant diversity were occurring at the AL site. The number of growing plant families decreased, but their species diversity increased (up to 25). The herbage was dominated by plants of the same families: *Asteraceae, Fabaceae*, and *Poaceae.* From 2015 to 2022, mouse-ear hawkweed (*Pilosella officinarum* F.W. Schultz et Sch. Bip.) and tall fescue (*Festuca arundinacea* Schreb.) spread greatly. The spread of reed fescue, which forms abundant biomass, in the soil may have influenced the increase in biomass yield during the last 10 years.

Correlational analysis confirmed that in stony sandy soil, as the duration of soil use increased, plant species diversity tended to decrease. In 27 years, the number of plant families decreased from 17 to 9 (r = 0.72), and the number of plant species decreased from 39 to 25 (r = 0.70).

## 4. Discussion

Soil organic carbon. The experiment confirmed the conclusions of various scientists regarding the positive influence of grasses on soil fertility [30,31,32] and enabled the determination of changes in their dynamics and differences depending on the use of grassland in *Arenosol.* In sandy and stony soil, SOC accumulation occurred more intensively at the AL site, where the herbaceous cover was formed from natural vegetation. Over 30 years, the SOC concentration increased by 23.5% (from 10.2 to 12.6 mg kg^−1^ SOC, *p* < 0.05), and its stocks in the Ah horizon by 27.8%. Other researchers [33] also note the positive effect of abandoned land vegetation on SOC sequestration. For 30 years, the thickness of the Ah horizon at the AL site also increased substantially (+4.3 cm, *p* < 0.05), which is partly related to the leaching of soluble organic carbon from the upper layer. This is a process characteristic of the Lithuanian climate; because the amount of precipitation exceeds the evaporation, a leaching moisture regime is formed, which results in the annual leaching of about 23 kg ha^−1^ SOC from the upper layer [34]. When the depth of the A-horizon increases, it is due to biological activity as well, particularly bioturbation processes, which play a dominant role in this transformation.

Thirty years after the conversion of arable land use to herbaceous land use, a statistically significant increase in SOC content and Ah horizon thickness occurred only at the AL site. At the MGf site, SOC content and Ah horizon thickness also increased, but the change was insignificant (*p* > 0.05). In the MGunf site, the variation in these indicators was the smallest and did not differ significantly from MGf. SOC accumulation slows down as the duration of use of the grassland increases. Similar data have been published by [35], who found in a long-term experiment that a faster SOC occurred in the 13–20th years of growth in natural grassland. They believe this is associated with higher above-ground and root biomass in this period, as well as with higher plant species diversity, especially C4 grasses and legumes. Ref. [36] also notes the positive influence of species diversity on grass productivity. They found that grass diversity promotes an increase in carbon and nitrogen stocks in the soil. Conversely, enhanced soil C and N stocks provide positive feedback to plant productivity through increased N mineralization, which could further accelerate soil C and N storage in the long term.

The slowdown of SOC accumulation rates can be linked not only to changes in grass biomass yield and its succession, but also to climate change, especially an increase in air temperature. In the region where the experiment was carried out, during the last decade, the air temperature often significantly exceeded the SKN and precipitation was scarce, stimulating the SOM mineralization processes. Scientific research has established that an increase in air temperature, and at the same time soil temperature, directly alters the structure of soil bacterial communities and could substantially alter their function [37,38]. It has been determined [17] that the decomposition of soil organic matter is positively correlated with temperature. Rocci at al. [39] after performing a meta-analysis of published data, concluded that different SOC compounds respond differently to climate warming. SOC changes in soil are best indicated by particulate organic carbon. It is also emphasized that different ecosystems are not equally resistant to SOM decomposition, so the effect of temperature must be analyzed for each region separately [40].

This experiment revealed that in sandy soil, AL increases C accumulation the most compared to MG land use, and the process itself is long term. Only after 30 years were significant changes in the amount of SOC, compared to its amount at the beginning of the research, detected [41]; summaries of long-term research publications determined that sites that were converted from cropland to grassland reached a SOC equilibrium level 47.3% above permanent cropland levels 83 years (95% CI: 79 to 90 years) after conversion. At the MGf site, the amount of SOC increased continuously (+10.0%) over 30 years, which was statistically insignificant. At the MGunf site, the amount of SOC changed slightly and remained similar to the levels observed at the beginning of the experiments. It should be noted that although the carbon content of the MGf soil increased, compared to the MGunf soil, it was insignificant even after 30 years. However, after evaluating the SOC accumulations in the entire Ah horizon, it became clear that they increased significantly (*p* < 0.05) in all land uses since the beginning of the research. SOC stocks increased the most in MGfert and AL soil, and less in MGunf, but the differences between land use types were statistically insignificant (*p* > 0.05). We believe that the effect of herbaceous plants on SOC sequestration is better reflected by the assessment of its amount in the entire layer of the Ah horizon, because the amount of SOC and the thickness of the Ah horizon itself change. It should be noted that in all studied land uses, the faster increase in SOC content remained comparable to the levels observed at the beginning of the experiments. It should be noted that in all studied land uses, a faster increase in SOC content took place in the first 20 years after the conversion of arable land to herbaceous land use.

Soil chemical properties. The changes in soil chemical properties (pH, available phosphorus, and potassium) were more contrasting in the grasslands of different uses. At the MGf site, using optimal rates of mineral fertilizers, calculated based on soil properties and plant needs, the amount of available phosphorus in the soil significantly increased over 30 years (+42 mg kg^−1^ P, *p* < 0.05), and the concentration of available potassium did not change significantly. Our research data are similar to the results of research conducted in Slovakia [42], which demonstrate that medium fertilization can be an acceptable compromise to meet both productivity and environmental aspects and to connect ecological benefits with social benefits in the long term. The average level of fertilization rates makes it possible to maintain soil properties, SOC content, and quality and to ensure good forage quality.

Without the use of fertilizers at the MG site, soil degradation occurs rapidly, and the amounts of available phosphorus and potassium significantly decrease. According to these indicators, the soil has become degraded and belongs to the group of low-fertility soils. An intense decrease in phosphorus and potassium occurred during the first 20 years. After reaching a concentration of 30–60 mg kg^−1^ P or K, the decrease in concentration slowed down. At the AL site, changes in available phosphorus and potassium were insignificant (*p* > 0.05), although the tendency of their concentration to decrease is apparent.

Soil acidity in different land uses also went in different directions. At the AL site, the pH indicator has not changed significantly, while in MG land use, the soil has become acidified by 0.9–1.2 units of pH since the beginning of the research. Faster acidification took place during the first 20 years, and then the pace slowed down. The soil acidified the fastest at the MGf site. This was influenced by mineral nitrogen fertilizers. Analogous results have also been published by [43], who found in a long-term experiment that N-containing inorganic fertilizers strongly acidified the soil, and over 120 years, the acidity became less than pH 4.5. Soil acidification due to the use of mineral nitrogen fertilizers has been confirmed by other scientists [44]. However, in soils rich in calcium carbonate, the negative effects of nitrogen fertilizers on acidity may not occur [45].

Grassland and abandoned land biomass yield. The biomass yield of grasslands had different changing trends as the period of their use increased. At the beginning of the experiment, when the soil properties were still the same in all plots, although the yield was higher in the fertilized plot of MG land use, significant differences with the unfertilized plot became apparent only after about 6–8 years, when the amounts of available phosphorus and potassium in the soil decreased. After 20 years of land use, when the amounts of available phosphorus and potassium decreased to 60–80 mg kg^−1^, the biomass yield in the MGunf plot decreased significantly (*p* < 0.05) compared to MGf and did not differ from the biomass yield at the AL site (*p* > 0.05). According to [46], nitrogen and phosphorus deficiencies limit grassland biomass production, as well as the efficiency of micronutrients. Mineral fertilizers have a more positive effect on biomass yield than combined nitrogen, phosphorus, and potassium fertilizers, while the pH indicator has a weakly pronounced positive effect [47]. A long-term lack of nutritional elements limits the biomass yield and can also influence the species composition of plants.

Managed grassland and abandoned land botanical composition. As the MG forage ages, it loses alfalfa content, which reduces the forage value of the forage. As early as the fourth–fifth years of cultivation, in MGf, *Poaceae* plants dominated the grass mixture and constituted up to 99% of the total biomass. The negative effect of mineral nitrogen fertilizers on the distribution of *Legume* plants in grasslands is also confirmed by other scientists [43,48,49]. In a site that is not fertilized with mineral fertilizers, alfalfa plants last longer, and in the fourth–fifth year of growth, they can make up to 28% of the biomass of all plants. To maintain the forage value of the grassland after the reduction in alfalfa, it is possible to renew the grassland by sowing legume seeds into the existing grassland. Increasing the amount of legumes in mixtures not only improves forage value but also promotes the accumulation of C and N in the soil and enhances the soil structure [50,51].

At the AL site, over 30 years, the diversity of plants changed not only depending on the age of the AL but also on the meteorological conditions of the vegetation period. During the entire research period, plants of three botanical families (*Asteraceae*, *Fabaceae*, and *Poaceae*) dominated the AL site. As the duration of AL age increased, plant diversity decreased. The number of plant families decreased from 17 to 9, and the number of plant species decreased from 39 to 25. Twenty years after the conversion of arable land to abandoned land, mouse-ear hawkweed (*Pilosella officinarum* F.W. Schultz et Sch. Bip.) began to spread on the site as well as tall fescue (*Festuca arundinacea* Schreb.). The distribution of these plants may have been influenced by drier and warmer growing seasons, as they are more drought tolerant than other plants [52,53,54]. These data illustrate that abandoned land can only partially support plant biodiversity, but external factors can negatively influence the distribution of plant species. However, according to [55], the spread of tall fescue (*Festuca arundinacea* Schreb.) in the soil promotes faster accumulation of SOC and increases the biomass yield of abandoned land.

## 5. Conclusions

Our results support that two land uses—AL and MGf—can be effectively applied to prevent *Arenosol* degradation. In AL land use, SOC sequestration takes place more efficiently compared to MGf, and the initial chemical properties of the soil (pH, available phosphorus and potassium content) change a little, but no economic benefits are obtained in these cases. The productivity of AL vegetation biomass increases with the age of land use, which can be associated with an increase in SOC content and climate warming, leading to a longer growing season. However, an increase in the age of the AL site does not necessarily ensure plant diversity, as climate change and warming may allow drought-resistant plant species to spread or dominate the succession, such as *Pilosella officinarum* F.W. Schultz et Sch. Bip. and *Festuca arundinacea* Schreb.

Planting fertilized managed grasslands in *Arenosol* increases SOC content more slowly, but it creates conditions for environmentally friendly agricultural practices and ensures socio-economic stability in areas with marginal soils. Monitoring of soil chemical properties is necessary in MGf land use to stabilize or improve the levels of nutritional elements and soil acidity. The use of MG for fodder production without fertilizing causes degradation of Arenosol (the amount of available phosphorus and potassium significantly decreases, and the soil becomes acidified), although the SOC content remains stable. As the nutrient elements in the soil decrease, the productivity of MGunf significantly decreases, and after 20 years, it was analogous to the productivity of the natural vegetation at the AL site. The ability of grasses of different land uses to sequester carbon in the soil is better reflected by SOC accumulations throughout the entire Ah horizon, rather than solely by changes in its amount in the upper soil layer, which may vary with the thickness over a long period of grassland use. Twenty-three years after converting arable land to herbaceous land use, SOC stocks across the entire Ah horizon increased substantially in all land uses (AL, MGf, MGunf).

Statistically significant changes in soil properties—such as an increase in SOC content in the 0–25 cm layer and an increase in Ah horizon thickness in the AL site), as well as change in the amount of available nutrients in the MG site—occurred more than 20 years after the conversion of arable land use to herbaceous land use.

## Figures and Tables

**Figure 1 plants-14-00166-f001:**
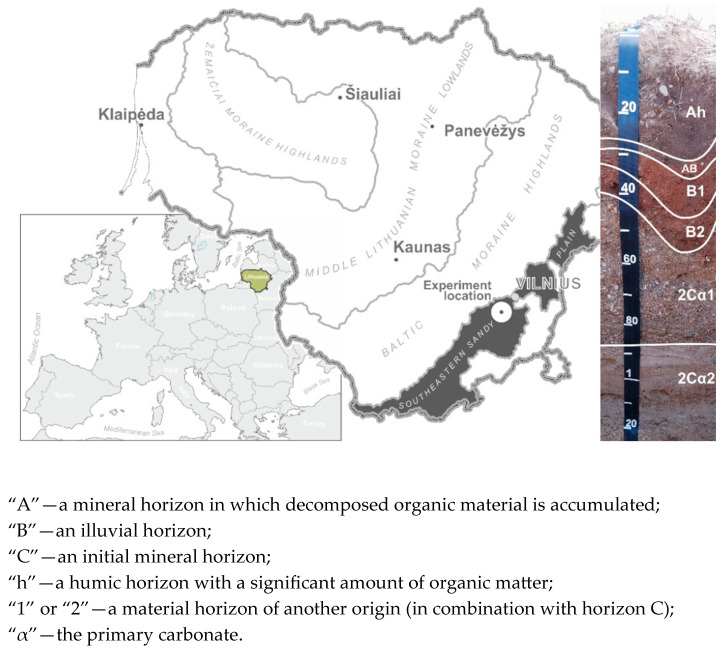
Study object and research site. Soil profile index according to WRB 2022.

**Figure 2 plants-14-00166-f002:**
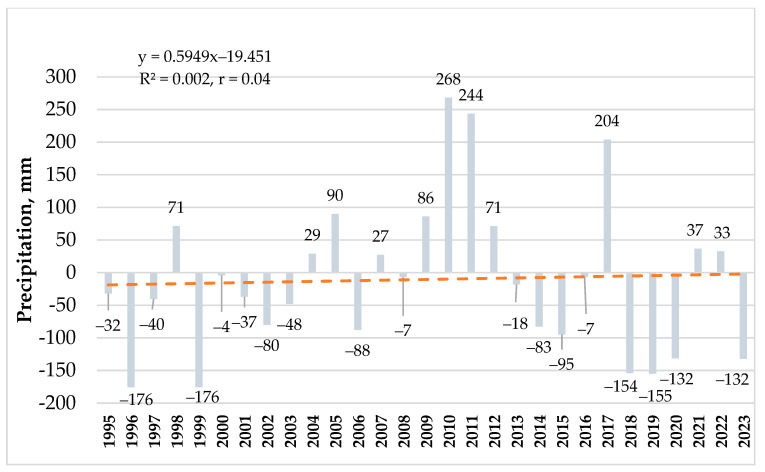
Deviation of annual precipitation from 1991 to 2020 compared to standard climate norms and its trend of change during the period of the experiment.

**Figure 3 plants-14-00166-f003:**
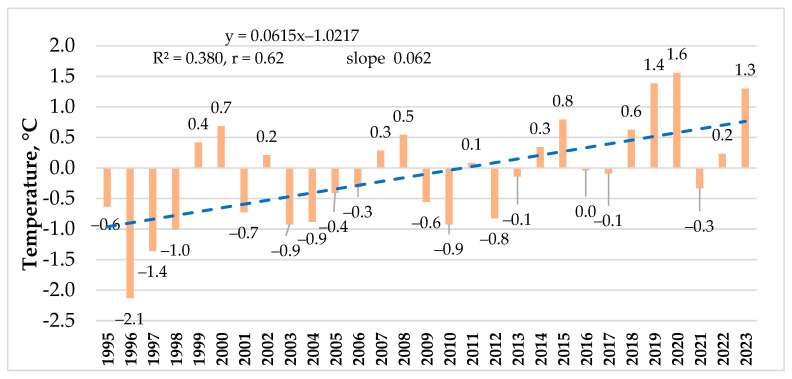
Average annual air temperature deviation from 1991 to 2020 compared to standard climate norms and its trend of change during the period of the experiment.

**Figure 4 plants-14-00166-f004:**
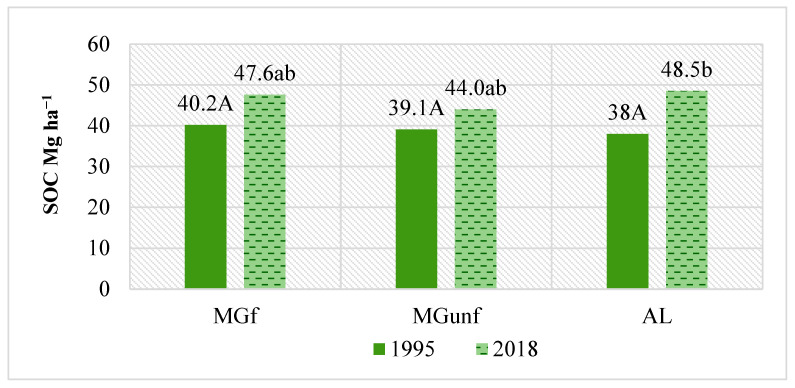
Sequestration of SOC (Mg ha^−1^) in the soil of grasslands with different uses in the Ah horizon. Note. Capital letters indicate a significant difference between treatments in 1995 (*p* < 0.05); lowercase letters indicate a significant difference between treatments in 2018 (*p* < 0.05).

**Figure 5 plants-14-00166-f005:**
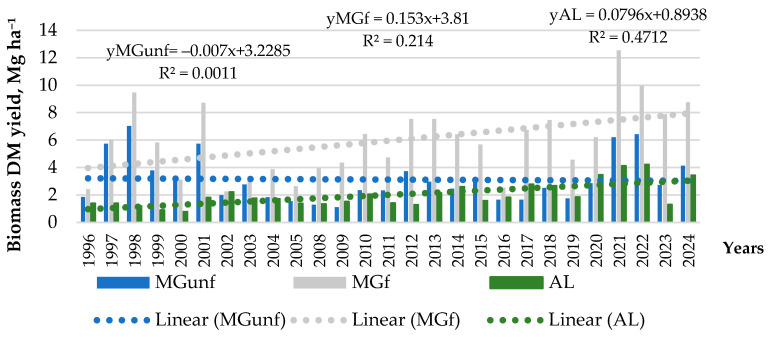
Biomass DM yield and its changing trends (Mg ha^−1^) for different grassland uses during the period of 1996–2024. Note. Abbreviations: AL, abandoned land; MGunf, managed unfertilized grassland; MGf, managed fertilized grassland.

**Figure 6 plants-14-00166-f006:**
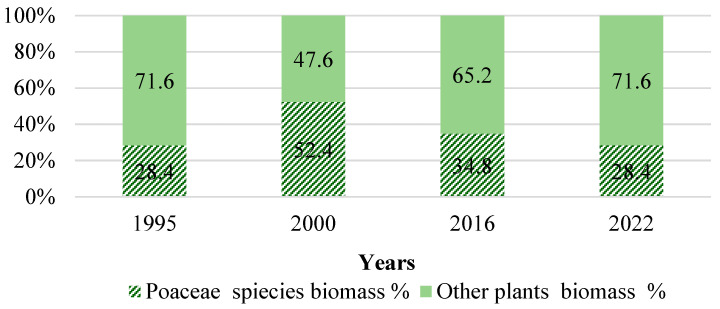
Relative amount (%) of *Poaceae* family plants in the biomass of the AL site.

**Figure 7 plants-14-00166-f007:**
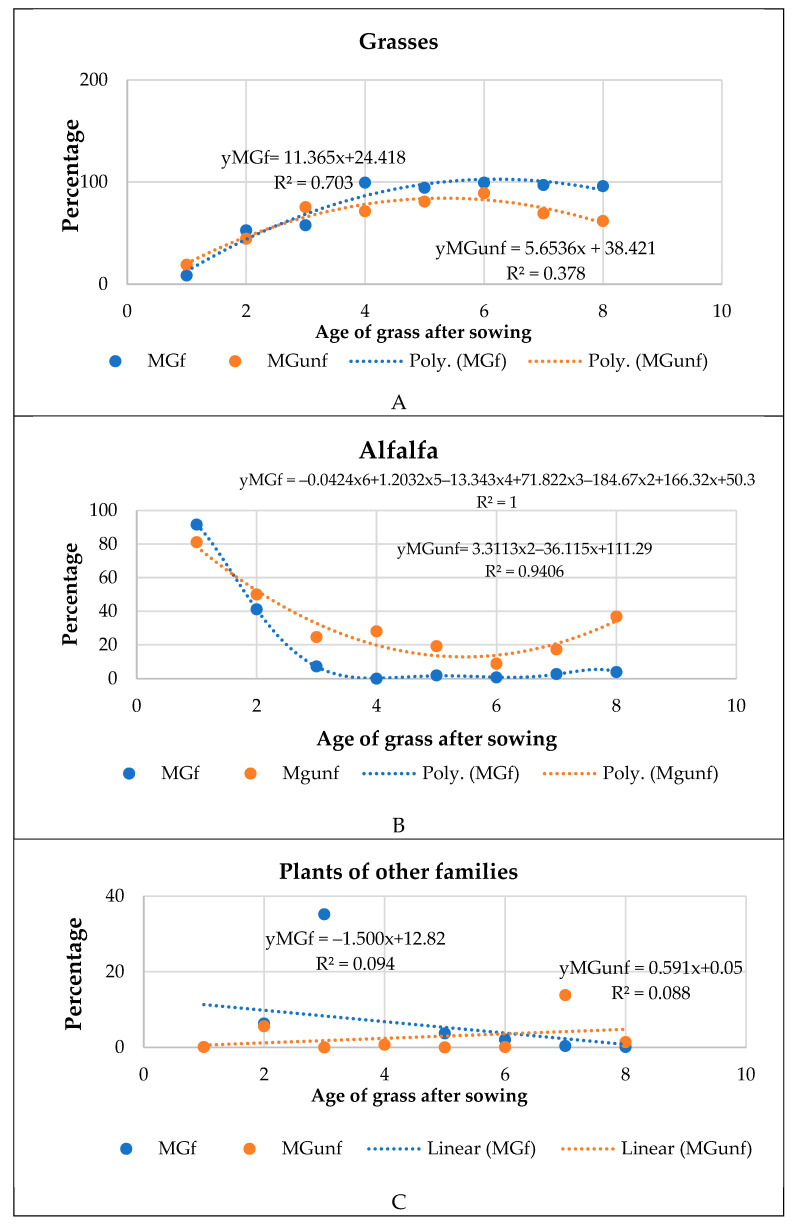
Variation in the botanical composition (**A**–**C**) of managed grassland during the year of cultivation. Note. Abbreviations: AL, abandoned land; MGunf, managed unfertilized grassland; MGf, managed fertilized grassland.

**Figure 8 plants-14-00166-f008:**
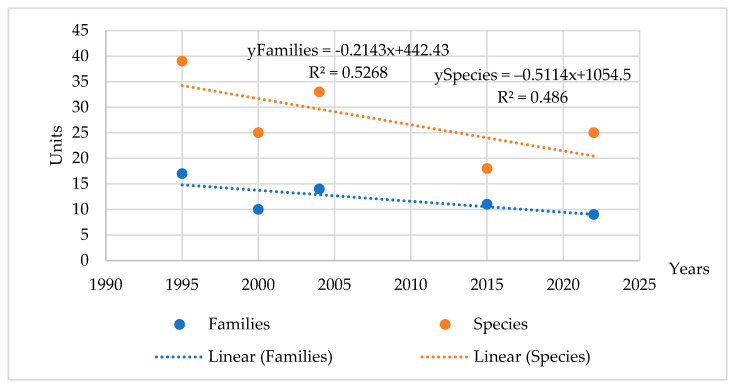
Changes in abandoned land succession over the course of the experiment (1995–2022). Note. Abbreviations: AL, abandoned land; MGunf, managed unfertilized grassland; MGf, managed fertilized grassland.

**Table 1 plants-14-00166-t001:** Soil characteristics of experimental site.

Horizon	Depth	Texture		
		Sand 63 μm–2 mm	Silt 2–63 μm	Clay < 2 μm
	cm	%		
Ah	0–28	88.58	7.67	3.75
AB	28–43	86.89	8.66	4.45
B1	43–70	84.55	7.95	7.50
B2	70–96	86.80	4.55	8.65
2Cα1	96–107	91.43	3.53	5.04
2Cα2	107–121	98.42	1.16	0.42

**Table 2 plants-14-00166-t002:** *Arenosol* chemical properties (0–25 cm).

Land Use	Measured Parameters	pH_KCl_	Available P mg kg^−1^	Available K mg kg^−1^
Managed Grassland Fertilized	1995	6.8 Bb	77 Aa	144 Ab
	2018	5.6 ABa	115 Cc	137 Cab
	2024	5.4 Aa	119 Bc	141 Cab
Managed Grassland Unfertilized	1995	6.8 Bb	77 Ab	144 Ab
	2018	6.0 Ba	34 Aa	58 Aa
	2024	5.9 ABCa	40 Aa	50 Aa
Abandoned land	1995	6.0 Aabc	69 Ab	141 Ab
	2018	5.8 ABa	66 Cab	120 Bab
	2024	6.1 Cc	61 Aab	121 Cab

Note. Capital letters in columns indicate a significant difference between soil indices in 1995, 2018, and 2024 (*p* < 0.05) in separate treatments, and lowercase letters in columns indicate a significant difference between the same treatment in 1995, 2018, and 2024 (*p* < 0.05).

**Table 3 plants-14-00166-t003:** The A horizon thickness and SOC content.

Land Use	Measured Parameters	A Horizon Thickness (cm)	SOC g kg^−1^
Managed Grassland Fertilized	1995	28.0 ab	9.9 abA
	2018	29.3 ab	10.0 abABC
	2024	30.3 b	11.0 bA
Managed Grassland Unfertilized	1995	28.0 ab	9.9 bA
	2018	29.0 ab	9.1 abA
	2024	29.3 b	9.8 abA
Abandoned land	1995	28.0 a	10.2 aAB
	2018	31.0 abc	12.1 abcC
	2024	32.3 c	12.6 cB

Note. Capital letters in columns indicate a significant difference between soil indices in 1995, 2015, and 2024 (*p* < 0.05) in separate treatments, and lowercase letters in columns indicate a significant difference between treatments in 2018 and 2024 (*p* < 0.05).

**Table 4 plants-14-00166-t004:** Mean values of grassland biomass (DM yield) and their variation indicators from 1996 to 2024.

Grassland Tipe	Average DM Yield Mg ha^−1^			Min	Max	VarS
	1996–2024	1996–2003	2017–2024			
MGf	5.95 c	5.11 cA	8.02 bB	2.25	12.55	6.84
MGunf	3.13 b	3.99 cB	3.53 aA	1.1	7.03	2.93
AL	2.01 a	1.43 aA	2.98 aB	0.78	4.22	0.79

Note. Abbreviations: AL, abandoned land; MGunf, managed unfertilized grassland; MGf, managed fertilized grassland. Capital letters in rows indicate a significant difference between DM yield in 1996–2003 and 2017–2024 periods (*p* < 0.05) in the same treatment, and lowercase letters in columns indicate a significant difference between different treatments in 1996–2024, 1996–2005, and 2015–2024 periods (*p* < 0.05).

## Data Availability

The original contributions presented in the study are included in the article; further inquiries can be directed to the corresponding author.

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
