# Peer review of "Perennial Grasses on Stony Sandy Loam *Arenosol*: Summary of Results of Long-Term Experiment in Northern Europe Region (1995–2024)"

_plants, 2025, doi:10.3390/plants14020166_

Round 1
Reviewer 1 Report
Comments and Suggestions for Authors
Dear authors, Dear Editor-in-Chief,
I am currently reviewing a manuscript titled "Perennial Grasses on Stony Sandy Loam Arenosol: Summary of the Results of a Long-term Experiment in Nord Europe region(1995-2024)", which is dealing with complex assessment of the long-term experiment (1995-2024) results carried out in Arenosols within a temperate climate zone with a special focus on productivity of natural and managed grasslands. I think this submission is of great importance for the research community and worth publishing in the journal. However, some revision will be needed to proceed to the next steps of production. Please consider some revision according to my comments:
1. Please consider changing the title to "Perennial Grasses on Stony Sandy Loam Arenosol: Summary of the Results of a Long-term Experiment in Northern Europe region(1995-2024)".
2. Please provide some illustrations on the study area - at least, the Map and some photos of the area of investigation.
3. Please also consider adding photo(s) of the soil profile investigated.
4. Please provide some Tables with Physico-chemical parameters of the soil investigated (lines 96 - 101 etc.)
5. Please improve the quality of the Figure 1. A. Deviation of annual precipitation from 1991-2020 standard climate norms during the 176 period of the experiment. In the current form it is not readable for the details. The same applies to Figure 2.
6. Please consider merging Results and Discussion Chapters into the one, since it would easen the readability of the manuscript throughout (in my opinion).
Overall, I think that the submission is of great importance and has potential to be published in the journal after some revision.
I wish authors all the best in reviewing the manuscript and quick production procedures further.
Kind regards,
reviewer
Please double-check after the revision
Author Response
Dear Reviewer,
Please accept the answers to the remarks.
Best Regards,
Asta Kazlauskaite-Jadzevice

Reviewer 2 Report
Comments and Suggestions for Authors
This manuscript is “Perennial Grasses on Stony Sandy Loam Arenosol: Summary of the Results of a Long-term Experiment in Nord Europe region (1995-2024)”. Comments are as follows.
(1) Abstract: The question or clear research objective addressed in this article is missing.
(2) Abstract: The research content has already been revealed by other researchers, so I did not see any innovative content.
(3) Abstract: “As the age of AL increased, plant biodiversity decreased, and drought-resistant plants began to spread.” The relevant reasons have not been revealed.
(4) The Keywords are more than five.
(5) Introduction: “Land use must prioritize the efficient production of biomass to achieve better economic, environmental, and social results. Our focus should shift toward integrated, systems-based approaches to land management that promote the sustainable intensification of agricultural production, even in areas such as neglected grasslands, abandoned lands. These areas share the common issue of having lost their economic and/or ecological viability for the community. This challenge is exacerbated by the ongoing degradation of such lands, which leads to a decline in their capacity to provide essential ecosystem services.” The description in this paragraph is not specific enough.
(6) Introduction: The mechanism of soil organic carbon sequestration lacks detailed introduction.
(7) Materials and Methods: The sampling and testing description of this study is not detailed enough.
(8) Materials and Methods: “This paper presents indicators of soil chemical properties in 1995, 2018 and 2024, which illustrate the dynamics of their change over a period of 30 years.” What are the reasons for choosing this time?
(9) Results: The analysis of deviations is missing in Tables 1 to 3 and Figures 1 to 5.
(10) Discussion: The discussion is divided into three parts (Soil organic carbon, Grassland and abandoned land biomass yield, Managed grassland and abandoned land botanical composition.), but it does not highlight the key innovative points.
(11) Conclusions: The discovery of innovation has not been found, and further summary is needed.
(12) The limitations of the research are lost.
(13) The format of references needs to be modified according to the requirements of the journal.
(14) Table 1 should not be distributed across two pages.
(15) The English language of this manuscript needs to be revised.
Comments on the Quality of English LanguageThe English language of this manuscript needs improvement.
Author Response

(The authors gave the same response as above.)

Reviewer 3 Report
Comments and Suggestions for Authors
This paper summarized the results of a long-term experiment (1995-2024) carried out in Arenosol within a temperate climate zone. The study focused on the productivity of natural and managed grasslands, their succession changes over time, the effects on soil chemical properties, and soil organic carbon (SOC) sequestration. The long-term experiment helps in understanding the SOC accumulation mechanisms. The manuscript can be published after minor revision.
1. Table 1. Replace the P2O5 and K2O with P and K, respectively.
2. Table 2. Change the title to "The A horizon thickness and SOC content".
3. Delete the frames of all Figures.
4. Figure 6. Nonlinear regression models can be used for Grasses.
5. The subfigures composing one Figure should be noted with different letters (e.g., a, b and c).
6. A substantial increase in the SOC stock in the abandoned land was observed from 1995 to 2018 in this study. Please discuss this point in detail.
7. Line 546. 0‒25 cm
Author Response

(The authors gave the same response as above.)

Reviewer 4 Report
Comments and Suggestions for Authors
The manuscript presents results from a long-term field experiment in Lithuania which was set up to study effects of land use change from arable land to grassland at marginal lands. At three points in time during a period of about 30 years soil conditions and plant development were investigated. A major focus was on the accumulation of organic carbon in the soil. Three management types were established. It turned out that both, fertilized managed grassland and abandoned land showed the highest carbon accumulation rates in the topsoil. At unfertilized grasslands, however, indicators for soil degradation by nutrient losses were found. Even if the positive effect of grassland on soil organic carbon accumulation is well-known, the presented results are certainly of interest for the scientific community. Particularly, the question of using marginal lands is of interest worldwide, so that this study may offer arguments for new management strategies for such sites.
In general, the manuscript is organized and written in a good manner. At some places minor linguistic deficiencies should be revised. A larger need for optimization can be found in the materials and methods description. The experimental setup is only vaguely described in the text. It would be helpful for readers to have maps, schemes of the different experimental variants or tables showing the exact treatment (e.g., fertilization, harvest etc.). Furthermore, the authors should think about the labelling of significances in the tables of the results section. It is a bit confusing and not completely clear to me which data sets are compared. In the explanation notes below the tables it is stated that the capital letters indicate differences between the years within the individual treatment. In that case it remains unclear, why, e.g., in Tab. 1 for variant “MGunf” in 2024 the letter combination “ABC” can occur.
Further remarks in detail:
l. 47/49: The content of these two sentences seems to be contractionary: First, it is stated that agrotechnical measures contribute to soil degradation. Then it is supposed that agrotechnical measures can improve soil properties.
l. 55: what do you mean with “lowest negative change”?
l. 58: The manuscript is focusing on marginal lands. The authors do not provide a general definition of this term. This seems to be of importance as the major concept of the paper is the change from arable lands to grasslands. A larger part of marginal lands, depending on the definition, is certainly not used as cropland, as soil quality is too low. Thus, the authors should clearly state in the introduction, which type of marginal land could benefit from the results of the study.
l. 75: Please explain why specifically Arenosols have been selected for this experiment.
l. 103: Is “abandonment” a type of land “use”? Was the AL variant also managed (e.g., to avoid the growth of woody plants)?
l. 136-139: Was the plant species cover only analyzed for AL? If it was also carried out for the other two variants, it would be good to have the results presented in a similar way. Furthermore, the data set should allow for calculating other, more informative indices for biodiversity. They should not only reflect the number of species (or families) but also their abundances (which should be collected using the Braun-Blanquet method).
l. 147: Doe the mentioned “core cutter samples” to the bulk density measurement? Results of the bulk density characterization are not shown in the results section. They are not only of interest for calculating carbon stocks, but can be used as an additional indicator of soil quality development. The correlation between bulk density and soil organic matter is well-established and could be tested here again.
Tab. 2: Lowercase letters: Did you only compare 2018 and 2024? Letters are also given for 1995.
Fig. 3: Indicators for standard deviation would be a necessary additional information. What about data for 2014?
Fig. 6: Title of the upper part: “Grasses” in contrast to “other grasses” (figure below)? Which species or families are meant here? Middle part: Why are the negative values above? Notes: “AL” is not shown here.
l. 398: The increasing depth of A-horizons is not only the results of leaching processes but mainly of biological activity. Bioturbation processes should be considered as one of the major reasons.
l. 464: What is the definition of “low fertility soils”? Is it related to certain threshold values for nutrients?
l. 520 (see also above): Is "AL" really a type of land “use”?
Author Response

(The authors gave the same response as above.)
